# The Ryanodine Receptor as a Sensor for Intracellular Environments in Muscles

**DOI:** 10.3390/ijms221910795

**Published:** 2021-10-06

**Authors:** Takuya Kobayashi, Nagomi Kurebayashi, Takashi Murayama

**Affiliations:** Department of Cellular and Molecular Pharmacology, Graduate School of Medicine, Juntendo University, Tokyo 113-8421, Japan; nagomik@juntendo.ac.jp

**Keywords:** ryanodine receptor, skeletal muscle, cardiac muscle, exercise and injury, heart function, diet

## Abstract

The ryanodine receptor (RyR) is a Ca^2+^ release channel in the sarcoplasmic reticulum of skeletal and cardiac muscles and plays a key role in excitation–contraction coupling. The activity of the RyR is regulated by the changes in the level of many intracellular factors, such as divalent cations (Ca^2+^ and Mg^2+^), nucleotides, associated proteins, and reactive oxygen species. Since these intracellular factors change depending on the condition of the muscle, e.g., exercise, fatigue, or disease states, the RyR channel activity will be altered accordingly. In this review, we describe how the RyR channel is regulated under various conditions and discuss the possibility that the RyR acts as a sensor for changes in the intracellular environments in muscles.

## 1. Introduction

Skeletal and cardiac muscles are essential organs for the maintenance of life, which embody most of the daily activities, such as walking, eating, and creating. Skeletal muscles are required to perform flexible and complex movements, generate and maintain force, and respond rapidly to different conditions, while the heart muscle sustains and regulates heartbeat according to physical activity. In addition, during human physical activities, muscles are subjected to high-loads and fatigue and various external stimuli, such as injury, stress, and excitement. These external stimuli cause intricate changes in the intracellular environment of muscle cells. The muscle can sense these intrinsic changes and immediately respond to them.

The contractile activity of skeletal and cardiac muscles is induced by a series of events, including action potentials, Ca^2+^ release from the sarcoplasmic reticulum (SR), and contractile force generation. The ryanodine receptor (RyR) is a Ca^2+^ release channel in the SR. In skeletal muscle, the type 1 RyR (RyR1) interacts with the dihydropyridine receptor (DHPR or Cav1.1), an L-type Ca^2+^ channel in the transverse tubule (T-tubule). As a result, the RyR1 channel opening is triggered by a conformational change in the DHPR after depolarization of the T-tubule (referred to as depolarization-induced Ca^2+^ release, DICR) [1,2,3]. In cardiac muscle, the type 2 RyR (RyR2) is opened by the binding of Ca^2+^ that is introduced from the extracellular space through the DHPR (Cav1.2) (referred to as Ca^2+^-induced Ca^2+^ release, CICR). However, in cardiac muscle, a direct interaction between the RyR2 and DHPR is lacking. The opening of RyR channels releases massive Ca^2+^ from the SR. Thus, the intracellular Ca^2+^ concentration rapidly increases from 50 nmol/L during the resting state to 10–20 μmol/L as a maximum peak in twitch contraction [4,5,6]. A structural change in tropomyosin on the thin muscle filament is caused by Ca^2+^ binding to troponin C, and this change facilitates the interaction between myosin and actin, myosin ATPase activity and the sliding of the two filaments, resulting in muscle contraction [7,8]. Thus, the RyR is one of the most important elements in excitation–contraction coupling and a central component for the myocyte function.

The RyR is a huge (>2 MDa) protein complex, which is composed of four 550 kDa subunits. The RyR is a six-transmembrane P-type channel at the C-terminus with a large N-terminal cytoplasmic region which covers 70% of the entire molecule [9]. Recent cryo-EM studies of RyRs revealed a complex structure with over 15 domains [10,11,12]. The transmembrane S1, S3 and S5 helices penetrate the SR membrane from the cytosolic side to the luminal side, whereas the S2, S4 and S6 helices penetrate the membrane in the opposite direction. The S6 helix forms a pore for Ca^2+^ ions. The S2-S3 domain, which is the region between S2 and S3 helices on the cytosolic side, is thought to be involved in regulation of channel opening. The S4–S5 linker is a short helix that runs parallel to the SR membrane between the S4 and S5 helices and participates as a stopper against the movement of S6 helix. The C-terminal domain (CTD) forms a part of Ca^2+^ and nucleotide binding sites at the C-terminal side of the S6 helix. The C-terminal domain is involved in channel gating and comprises the channel core. The large N-terminal cytoplasmic region which is unique to the RyR family contains multiple domains, including the N-terminal domain (NTD), SPRY1, P1, SPRY2, SPRY3, Handle, HD1, P2, HD2, and CENTRAL domains (Figure 1). Since many disease-linked point mutations have been identified in RyR cytoplasmic domains, this region is thought to be involved in the regulation of channel activity [13].

In this review, we will first describe the regulatory factors, such as small molecules, proteins, and modifications that affect RyR channel activity from a biochemical perspective (Table 1). Next, we will discuss changes in these regulatory factors in muscles during daily activities and crisis situations to understand macroscopic biological phenomena from a molecular perspective. In this way, we will propose RyRs as intracellular environmental sensors for various signals.

## 2. Factors Affecting RyR Function

### 2.1. Ca^2+^

Ca^2+^ is a primary ligand for RyRs that induces channel opening. The channel activity of all RyR subtypes increases at micromolar concentrations of Ca^2+^. However, Ca^2+^ does not only open the channel but also leads to channel closing when higher concentrations are reached (sub-millimolar or more) [14,15]. The biphasic Ca^2+^ dependence is explained by the presence of two distinct Ca^2+^-binding sites: a high-affinity site involved in opening the RyR channel and a low-affinity site for inactivation [16,17]. The Ca^2+^ concentration for the inactivation differs between RyR1 and RyR2; the activity of RyR1 is inhibited by sub-millimolar Ca^2+^; however, millimolar or more Ca^2+^ level is required for the inhibition of RyR2. The high-affinity Ca^2+^-binding site for channel activation is located in the interface between the Central domain and CTD [10] (Figure 1).

### 2.2. Mg^2+^ and Adenine Nucleotides

Mg^2+^ has an inhibitory effect on the RyR channels [17,18,19]. Mg^2+^ acts as a competitive inhibitor for the activating high-affinity Ca^2+^-binding site and an agonist for the inactivating, low-affinity Ca^2+^ site [20,21]. The EC_50_ of Mg^2+^ for the inactivating Ca^2+^-binding site is similar to that of Ca^2+^, whereas the EC_50_ is lower than that of Ca^2+^ for the activating Ca^2+^ site [22,23]. Although Mg^2+^ is predicted to bind to these Ca^2+^-binding sites, there is no structural evidence for Mg^2+^ binding thus far.

Adenine nucleotides (ATP, ADP and AMP) increase the RyR activity [18,19]. The rank order of activation is ATP>ADP>AMP. The EC_50_ of ADP to RyR2 is one order of magnitude higher than that of ATP [24]. The concentration of ATP, ADP and AMP at resting skeletal muscle is 8.5 mmol/L, 0.008 mmol/L and 0.007 μmol/L, whereas those after exercise is 8.5 mmol/L, 0.15 mmol/L and 2.7 μmol/L, respectively [25]. RyR is predicted to have two ATP-binding sites in one protomer [26], but structural studies have revealed that only one ATP is identified at the interface between Central, S6 and CTD which is near the Ca^2+^ binding site [10] (Figure 1). Because other adenine nucleotides can also bind to the site, displacement of ATP by ADP might occur when ADP concentration is increased. In consequence, an increase in ADP concentration causes a reduction in Ca^2+^ release.

The variation in the concentration of the factors described above is relevant to RyR activity in high-load exercise. We will discuss these changes further in the *Exercise and fatigue* section of this review.

### 2.3. SR Luminal Ca^2+^ and Calsequestrin

Ca^2+^ release through RyR2 occurs spontaneously when a high luminal Ca^2+^ concentration is present [27,28,29]. Some studies have discussed the mechanism of sensing the luminal Ca^2+^ concentrations in relation to ATP. Single-channel recording showed that the affinity of the RyR for ATP decreased when the concentration of luminal Ca^2+^ was less than 1 mM, and the effect of ATP on RyR activation increased when the concentration of luminal Ca^2+^ was 8–35 mmol/L [30]. Single-channel recording experiments showed that the activity of RyRs increased when the concentration of luminal Ca^2+^ increased above 1 mmol/L in the SR, whereas trypsin treatment of the luminal side of the SR membrane decreased RyR activity after Ca^2+^ concentrations increased [31]. Therefore, it was proposed that RyRs have at least two types of luminal Ca^2+^ sensor sites in the SR lumen: one for activation and one for inhibition. A recent study of RyR2 showed that the E4872A mutation abolished luminal Ca^2+^ activation, suggesting that this site may be part of the luminal Ca^2+^ sensor [32]. It was predicted by simulation that Ca-binding with the luminal region of RyR was also predicted by simulation study [33]. In imaging studies of flexer digitorum brevis [34] and tibialis anterior [35] muscles, the concentration of luminal Ca^2+^ was ~0.4 mmol/L at rest and fell to ~0.08 mmol/L after tetanic stimulation at 50 Hz.

Calsequestrin is a low affinity, high capacity Ca^2+^-binding protein in the SR and is thought to act as a Ca^2+^ buffer [36,37,38]. In addition, calsequestrin inhibits RyR channel activity [39,40]. It has been reported that the inhibitory effect of calsequestrin is induced when the luminal Ca^2+^ concentration drops below 0.1 mmol/L [41,42]. Moreover, calsequestrin binds to junctin and triadin, which are SR membrane proteins that promote Ca^2+^ release though the RyR, and depresses the effect of these proteins [42,43]. Conversely, calsequestrin dissociates from triadin with the increasing the luminal Ca^2+^ concentration, and inhibitory effect of calsequestrin on RyR is reduced [39,44]. The variation in the concentration of the factors described above is relevant to RyR activity in high-load exercises. We will discuss them in the *Exercise and*
*Fatigue* section of this review.

### 2.4. Dihydropyridine Receptor (DHPR)

In skeletal muscle, RyR1 is under the control of the DHPR. Therefore, abnormalities in the DHPR are directly related to RyR1 function [45,46,47]. Amino acid mutations in the DHPR cause diseases such as malignant hyperthermia (MH) and myopathy [48]. Malignant hyperthermia is a life-threatening disorder characterized by skeletal muscle rigidity and elevated body temperature in response to volatile anesthetics [49]. Although the primary genetic causes of MH are mutations in the RYR1 gene [50], some mutations have also been found in the genes for α_1S_ and β_1a_ subunits of DHPR [48].

### 2.5. FKBP12

FKBP12 is a protein that binds to the immunosuppressive agent FK506, which is used in patients with organ transplants or autoimmune diseases. FKBP12 binds to RyRs [51] and modulates their activity [52,53,54]. FKBP12.6 is an isoform of FKBP12 with a different molecular weight and is expressed abundantly in the heart. RyR1 and RyR2 tightly bind to FKBP12 and FKBP12.6, respectively [52,53,54]. Because of the tight binding between the RyRs and FKBP12/FKBP12.6 (K_D_ ~nmol/L), the RyR channel is thought to form an octamer (4 RyR subunits and 4 FKBP12/12.6 proteins). FKBP12 has an inhibitory effect on RyR1 activity [55]. In addition, FKBP12 may promote binding between RyR1 and the DHPR [56]. Thus, RyR1 activity is constantly suppressed by FKBP12. Similar effects were proposed for RyR2 and FKBP12.6 [57]. However, it has been reported that FKBP12 has no inhibitory effect on the RyR in the aorta [58]. The variation in the concentration of the factors described above is relevant to RyR activity in exercise, oxidative stress and ischemia/reperfusion injury. We will discuss them in the *Exercise and*
*Fatigue,*
*Skeletal*
*Muscle*
*Injury* and *Modulation of*
*Heart*
*Function in*
*Health and*
*Disease* section of this review.

### 2.6. Calmodulin

Both RyR1 and RyR2 can bind to calmodulin [59]. Calmodulin has biphasic effects on RyR1 depending on whether Ca^2+^ is bound to the protein. Apo-calmodulin (Ca^2+^ free) activates the RyR1 channel, whereas Ca^2+^-calmodulin inhibits it [60,61,62]. Ca^2+^-calmodulin inhibits RyR2 activity, but apo-calmodulin has no effect [63]. The complex structures of the RyR and calmodulin have been determined, and it was shown that calmodulin binds to the Central domain of the RyR [64,65]. S100A1, a 21 kDa Ca^2+^-binding protein, can also bind to the calmodulin binding site of the RyR, and it has been reported that the peak of Ca^2+^ transients is reduced in skeletal muscle lacking this protein [66]. We will discuss this change in the *Modulation of Heart Function in Health and Disease* section of this review.

### 2.7. Phosphorylation

Phosphorylation is an important modification in intracellular signal transduction. It has been demonstrated that the RyR is phosphorylated [67]. Although the mechanism is not fully understood, phosphorylation may alter the activation of the RyR that is mediated by ATP [68]. In RyR2, there are currently three serine residues that are targets for phosphorylation (S2030, S2808, and S2814). These amino acids are thought to be phosphorylated by protein kinase A (PKA), protein kinase C (PKC) or Ca^2+^/calmodulin-dependent protein kinase II (CaMKII) [69]. The variation in the concentration of the factors described above is relevant to RyR activity in exercise, ischemia/reperfusion injury, catecholaminergic polymorphic ventricular tachycardia (CPVT) and diabetes. We will discuss these factors in the *Modulation of Heart Function in Health and Disease and Diet and Diabetes* sections of this review.

### 2.8. S-Nitrosylation and Oxidation

There are thiols in the RyR that undergo oxidative modifications [70], and the channel activity increases when these thiol groups undergo S-nitrosylation or S-glutathionylation. In experiments using various nitric oxide (NO) donors, some donors enhanced activity, while others showed an inhibitory tendency [71]. These results suggest that there may be multiple target cysteine residues that undergo oxidative modification [72]. The RyR1 Cys3635 residue has been identified as an S-nitrosylation target [73]. It has been reported that thiol modifications promote not only the activity of the RyR itself but also the dissociation of FKBP12 and Ca^2+^-bound calmodulin [74], thereby leading to an increase in channel activity by relieving FKBP12-mediated suppression (Figure 2A).

It is also known that RyRs can be oxidized by reactive oxygen species (ROS), especially hydroxyl radicals (•OH), which dissociate calmodulin from RyR2 and release the receptor from calmodulin-induced inhibition [75,76]. Hydroxyl radicals are produced from H_2_O_2_ by the Fenton reaction: therefore, the concentration of •OH is closely related to that of H_2_O_2_. H_2_O_2_ is generated in mitochondria and easily permeates the cell membrane. The production of H_2_O_2_ in normal tissues has been reported to be 100–400 pmol/min/mg [77,78,79,80]. In biochemical experiments, H_2_O_2_-induced RyR activation occurs at concentrations below 1 mmol/L, while higher concentrations cause inhibition [81]. The inhibitory effect is abrogated by thiol modification, and it has been reported that S-nitrosylation protects the RyR from oxidation by ROS [82]. Oxidative modifications occur not only on the RyR but also on FKBP12.6, which causes dissociation from RyR2 and relieves the receptor from FKBP12.6-mediated suppression [83,84]. Furthermore, the treatment of RyR2 with dithiothreitol altered the activity of RyR2 in response to luminal Ca^2+^ concentration [85]. Therefore, it is possible that cytoplasmic redox status affects the RyR2 Ca^2+^ sensitivity as well [86]. The variation in the concentration of the factors described above is relevant to RyR activity in high load-bearing work, muscle injury, heat stress, and individuals who consume a high-fat diet. We will discuss these changes in the *Exercise and Fatigue, Skeletal Muscle Injury, Modulation of Heart Function in Health and Disease* and *Diet and Diabetes* sections of this review.

### 2.9. NADH/NAD

Nicotinamide adenine dinucleotide (NADH) is a cofactor in redox reactions, and the ratio of NADH to NAD changes in cells during energy metabolism. NADH and NAD have been shown to increase RyR1 activity [87]. The NADH was reported as 0.1–0.5 mmol/L in resting skeletal muscle [88,89], and the ratio of NADH/NAD was 0.002–0.003 [90,91]. The EC_50_ of NADH and NAD for activation of RyR1 was shown to be 0.23 mmol/L and 0.31 mmol/L, respectively [87]. For RyR2, activity is decreased by NADH, and NAD counteracts the effect of NADH and restores the RyR2 channel activity [92]. Because the effects of NADH and NAD are opposite to each other, the ratio of NADH/NAD is a key factor in heart function. The ratio of NADH/NAD was reported to be 0.04–0.05 in heart muscle [93,94,95]. The variation in the concentration of the factors described above is relevant to RyR activity in myocardial metabolism. We will discuss this change in the *Modulation of*
*Heart*
*Function in*
*Health* and *Disease* section of this review.

### 2.10. Acyl-CoA

Acyl-CoA [96] and acyl-carnitine [97], which are metabolic intermediates of lipid metabolism, have been shown to directly increase RyR activity. In particular, palmitoyl-CoA is thought to reduce RyR inhibition mediated by Mg^2+^ [98]. Furthermore, acyl-CoA and acyl-CoA-binding protein may interact with FKBPs to regulate RyR activity [99]. The variation in the concentration of the factors described above is relevant to RyR activity in high-fat diet. We will discuss these factors further in the *Diet and Diabetes* section of this review.

## 3. RyR Regulation under Different Muscle Conditions

### 3.1. Exercise and Fatigue

Exercise and high load-bearing work cause various changes in skeletal muscle. In particular, during eccentric contraction, the expression of the NADPH oxidase (Nox)4, which produces superoxide from oxygen is increased and substantial ROS are generated [100,101]. Nox inhibitors abolish the tetanic stimulation-induced increase in Ca^2+^ release in skeletal muscle myotube [102]. This result might indicate that exercise induced-ROS generation is involved in the acute activation of RyR1 and enhancement of muscle contraction, whereas RyR1 function is impaired if ROS continued to oxidize RyR1 [81]. It has also been reported that the binding of FKBP12 to RyRs in muscle is reduced by eccentric contraction [103], and as noted above, dissociation reduces FKBP12-mediated suppression of RyR activity (Figure 2A). In most cases, exercise increases RyR activity, which in turn increases muscle tension.

When high-frequency stimulation is sustained, muscle tension gradually declines and the muscle falls into a state of fatigue. Under these conditions, a large amount of ATP is consumed by myosin and sarcoendoplasmic reticulum calcium ATPase (SERCA) in the muscle. Studies measuring ATP concentrations in creatine-kinase inhibited muscle fibers have shown that the ATP concentration in resting muscle was estimated to at 6–8 mmol/L, but decreased to approximately 3 mmol/L during contraction and 1.7 mmol/L when consumption was high [104]. Experiments using skinned fiber have shown that contractility decreased when the ATP concentration fell below 2 mmol/L [105]. The concentration of Mg^2+^, which was released from MgATP also increased with the consumption of ATP [106]. However, normal exercise does not substantially reduce the ATP concentration in muscles [25,107,108], but rather a large amount of ADP is generated as a by-product [108]. ADP does not maintain RyR activity as efficiently as ATP [24]. Therefore, the abundant increase in ADP may suppress channel activity. When high load-bearing exercise increases the ADP/ATP ratio, the impact of ADP may be enhanced (Figure 2B).

The SR luminal Ca^2+^ concentration decreased from 1 mmol/L to 0.08 mmol/L after exercise and high load-bearing work in one study [34]. The RyR senses a reduction in luminal Ca^2+^ concentration and channel activity decreases [30]. In addition, when luminal Ca^2+^ is depleted, calsequestrin binds to the RyR and suppresses its activity. Moreover, the acidification of the muscle cytosol by exercise [25] causes the suppression of RyR activity. Thus, high load-bearing work results in muscle fatigue that suppresses RyR activity (Figure 2B).

### 3.2. Skeletal Muscle Injury

Skeletal muscles are close to the body surface and are sensitive to external environments. Therefore, strong pressure or blows can lead to muscle contusions. It is known that muscle injury increases the production of H_2_O_2_ [109] and NO [110,111] (Figure 2C). Although ROS, including H_2_O_2_ and NO are known to increase RyR activity, an increase in RyR activity does not occur in damaged muscle fibers during exercise because damaged fibers are unable to contract [110,112]. However, it has been shown that ROS can spread from the site of injury and affect other muscle fibers and tissues. In particular, H_2_O_2_ has a large impact because it can easily permeate the cell membrane. Excessive contraction of the surrounding fibers because of increased RyR activity may increase damage in the contusion site. It has been reported that cooling of the contusion site reduces ROS production [112]. Thus, preventing the increase in RyR activity after muscle injury may be an important aspect of early treatment.

### 3.3. Stresess and Aging

Heat stress and hypoxia at high altitude also affect skeletal muscle. Heat stress is known to cause a transient increase in oxidation [113], to stimulate mitochondrial ROS production [114], and to promote the expression of endothelial nitric oxide synthase (NOS) [115] (Figure 2C). It has been reported that mice with a malignant hyperthermia mutation in RyR1 display increased RyR activity and rapid fever after the induction of heat stress, with the unfortunate enhancement of mitochondrial ROS production, resulting in sudden death [116].

Hypoxia at high altitude changes redox state. Under this condition, NOS is induced by the hypoxia-induced transcription factor HIF-1a [117] and ROS-production increases with anearobic ATP synthesis; therefore, proteins are oxidized [118] and RyR1 binding by FKBP12 is relieved due to the increased ROS and NO [119] (Figure 2C). At high altitudes, muscle injury may be induced by intense contractions because the activity of RyR1 is already increased.

Aging must also be discussed as one of the critical conditions affecting the muscle. Although many studies have been performed on aging and muscle atrophy, there are not many reports on factors that regulate RyR activity. However, it was demonstrated that calsequestrin expression increases with age [120] (Figure 2D). Additionally, the coupling of the DHPR and RyR1 is also decreased with aging [121].

### 3.4. Modulation of Heart Function in Health and Disease

The heartbeat is caused by regular Ca^2+^ release from the RyR2 channel, which is controlled by an action potential originating at the plasma membrane. The rate of the heartbeat is usually 60–100 beats per minute in humans, but it is known to be higher during exercise and psychological stress. When the sympathetic nervous system is stimulated by exercise or excitement, β-adrenergic signaling activates PKA in the myocardium and increases amplitude of cardiac action-potential-induced Ca^2+^ release from RyR2 by direct and indirect mechanisms. The factors that enhance RyR2 activity are elevation of SR luminal Ca^2+^, which results from an increase in Ca^2+^ influx via L-type Ca^2+^ channels and enhanced SERCA activity, and increased phosphorylation of RyR2 (S2030 and S2808) [122,123] (Figure 3A). NO production is also increased under exercise and stress conditions. These factors all potentiate RyR2 activity and result in strong contractions [124,125].

Heart failure is a highly prevalent progressive cardiac disorder with high morbidity and mortality. It arises from various causes, including coronary artery disease leading to cardiac arrest, high blood pressure, atrial fibrillation, valvular heart disease, and cardiomyopathy. When the ejection force is reduced during heart failure, the heart is unable to sufficiently maintain blood flow to meet the oxygen demands of tissues. The sympathetic nervous system is activated in the early stage of heart failure, which promotes PKA phosphorylation of RyR2 and NO production [126] (Figure 3B). It was reported that inducible NOS expression increased in failing cardiomyocytes [127]. In addition to the increase in NOS, an increase in Nox expression has also been reported during heart failure [128,129]. Nox is an enzyme involved in the generation of ROS [130,131], and ROS promotes RyR oxidation and contributes to increased channel activity (Figure 3B). As noted above, phosphorylation, S-nitrosylation, and oxidation may contribute to the activation of RyR2 (Figure 3B). Moreover, it has been reported that FKBP12.6 dissociates from RyR2 as heart failure progresses [132,133], which may further contribute to increased RyR2 activity (Figure 3A). The immediate activation of RyR2 by early compensatory mechanisms that involve these factors may restore cardiac function and ensure the survival of individuals.

The chronic activation of RyR2 has deleterious effects on cardiac function since it causes diastolic Ca^2+^ leakage from the SR and generation of abnormal Ca^2+^ waves. The former leads to contractile dysfunction and the latter causes arrhythmias. Elevated intracellular calcium levels activate CaMKII, which is also involved in the phosphorylation of RyRs. CaMKII was found to be increased in failing cardiomyocytes [134] and may be involved in the subsequent development of disease-related ventricular remodeling [135] or arrhythmia [136]. Therefore, the excessive activation of RyR2 may lead to the worsening of heart failure rather than improvement of cardiac function. Interestingly, the introduction of calmodulin that had high binding affinity for RyR2 prevented the progression to arrythmia [137] and reduced the symptoms of CPVT in the animal model [138]. These results indicate that it may be desirable to return to a moderate level of RyR2 activity in the heart after the initial emergency period.

Similar to heart failure, impaired cardiac function caused by ischemia-reperfusion injury also promotes ROS and NO production, phosphorylation, and dissociation of FKBP12.6 from RyR2 [139]. Ischemia-reperfusion is defined as the restoration of coronary blood flow after an ischemic episode. This injury to the myocardium causes the reduction in ejection force, and the increased demand for oxygen in tissues is similar to that observed for intense exercise and heart failure. It has been reported that the phosphorylation of S2814 by CaMKII is enhanced after ischemia-reperfusion [140]. Increased oxidation of RyR2 is a result of the increase in ROS generation in cardiomyocytes [141,142]. The influence of ROS reaches the cardiomyocytes that surround the injured myocytes. Furthermore, ROS produced from tissues other than heart tissue are also involved in RyR2 modulation. Strong oxidative stress responses occur after burns and include the activation of RyR2 and leakage of calcium from the SR in cardiomyocytes, which leads to heart failure [143].

Heart failure has been shown to have a significant effect on skeletal muscle. Biopsies of skeletal muscle from patients with heart failure have shown that RyR1 binding to FKBP12 is reduced [144]. It has been reported that phosphorylation of the S2843 residue of RyR1 by PKA dissociates FKBP12 from the receptor, and phosphorylation and dissociation of FKBP12 were also observed in an animal model of heart failure [145] (Figure 3B).

### 3.5. Diet and Diabetes

RyRs may also be sensitive to the effects of diet. It has been shown that a high-fat diet enhances β-oxidation, which in turn enhances ROS generation in the mitochondria [146], which increases RyR1 activity. Prolonged S-nitrosylation, however, leads to a state of sustained high Ca^2+^ concentration, which activates CaMKII and enhances RyR1 phosphorylation [147]. Since increased cytosolic levels of acyl-CoA and acyl-carnitine increase RyR1 activity [96,97,98], a high-fat diet also acts to increase RyR1 activity (Figure 4A).

A high-fat diet also affects the myocardium. Accumulation of palmitoyl-carnitine, a long-chain fatty ester of acyl-carnitine, has been shown to increase mitochondrial ROS production, RyR2 oxidation and S-nitrosylation and FKBP12.6 dissociation, which in turn may lead to increased Ca^2+^ spark frequency and development of tachycardia [148] (Figure 4A). Long-chain unsaturated fatty acids, such as eicosapentaenoic acid (EPA), docosahexaenoic acid (DHA), and oleic acid, in contrast, have been shown to decrease the activity of RyR2 [149,150] (Figure 4B). Although the mechanism is not fully understood, DHA has been shown to enhance the stability of RyR2-FKBP12.6 binding through its oxidized neuroprostane by-product (NeuroP) [150]. Furthermore, DHA has been found to reduce RyR2 mRNA expression [151].

It has been reported that some heart diseases such as long-QT syndrome [152], nocturnal ventricular arrhythmias [153] or heart failures [154] are caused by diabetes mellitus. The relations between diabetes and sympathetic system have been discussed as follows: the cardiac sympathetic afferent reflex is enhanced in early stage of diabetes [155], and the change of heart rate is caused in diabetes patients [154]. It has been known that the activation of sympathetic nervous system accelerates phosphorylation of RyR2 by PKA (Figure 4C). The hyper-phosphorylation of RyR2 is shown in heart of diabetes [156,157,158]. The increase in oxidation [156,159,160] and the decrease in FKBP12.6-binding RyR2 [157,158] was also observed, which are involved in activation of RyR2 (Figure 4C). This scheme is similar to the compensatory mechanism occurring in heart failure progress.

## 4. Conclusions

The RyR should not be considered simply as an ion channel, but also as a sensor and receptor that actively regulates skeletal and cardiac muscle function. When skeletal muscle is subjected to high-load exercise, the channel activity becomes reduced by alteration of nucleotide composition. After the occurrence of a critical situation, such as external stress, injury, or excitement of skeletal or cardiac muscle, changes in redox state and dissociation of inhibitory factors increase channel activity. Since RyRs are submaximally active under normal muscle conditions, they can sense changes in the intracellular environment caused by external influences and perform biphasic modulation of Ca^2+^ release. The balance of RyR channel activity and biphasic modulation associated with the intracellular environment may be crucial to maintain muscle function, and an imbalance in RyR activity may lead to skeletal and cardiac muscle disease.

## Figures and Tables

**Figure 1 ijms-22-10795-f001:**
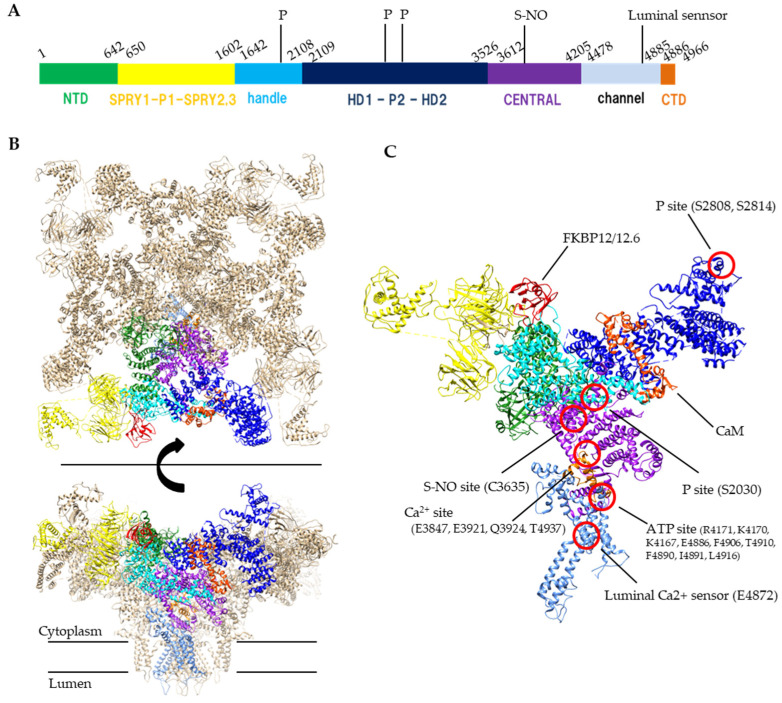
Structure of RyR. (**A**) A schematic illustration of domain organization of mouse RyR2. (**B**) Structure of the tetrameric RyR2 in complex with FKBP12.6 and calmodulin, looking from the cytoplasmic side (top) or parallel to the membrane (bottom). PDB code: 6JI8. (**C**) Structural details of domain organization and modification sites in one protomer. S-NO site and p site indicates s-nitrosylation site and phosphorylation sites, respectively. Domain colors correspond to a schematic illustration of A, NTD (residues 1–642) colored forest green, SPRY1-P1-SPRY2-SPRY3 (residues 650–1602) colored yellow, handle (residues 1642–2108) colored cyan, HD1-P2-HD2 (residues 2109–3526) colored blue, CENTRAL (residues 3612–4205) colored purple, channel region (residues 4478–4885) colored light blue, CTD (residues 4886–4966) colored orange. FKBP12.6 and calmodulin (CaM) are colored red and orange red, respectively.

**Figure 2 ijms-22-10795-f002:**
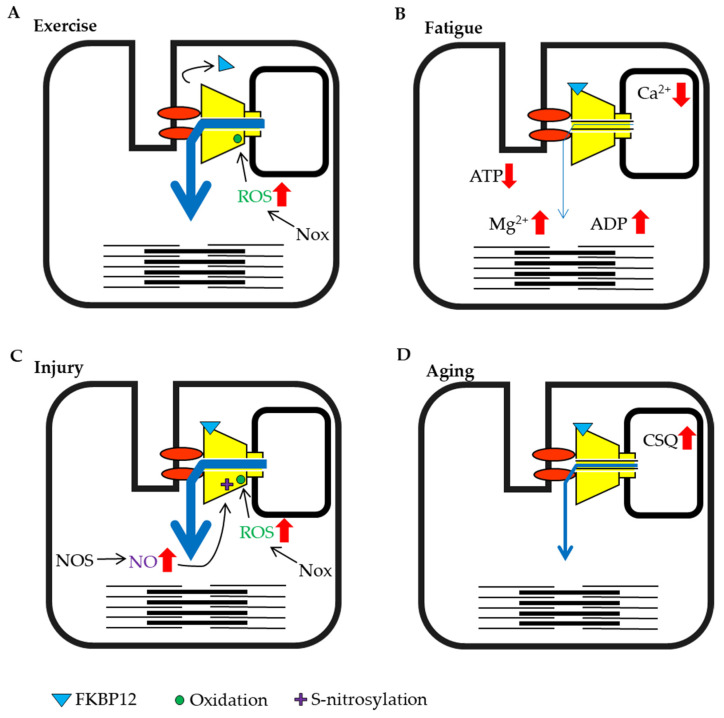
Schematic representation of skeletal muscle ryanodine receptor (RyR1) modulation by intracellular factors under different conditions. (**A**) Exercise training by eccentric contraction. (**B**) High load bearing work and fatigue. (**C**) Injury condition caused by damage, heat stress or disuse. (**D**) Aging.

**Figure 3 ijms-22-10795-f003:**
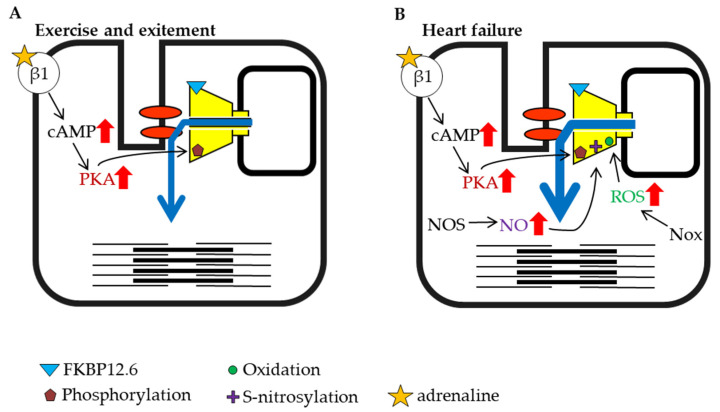
Schematic representation of cardiac ryanodine receptor (RyR2) modulation by intracellular factors under different conditions (part 1). (**A**) Exercise and excitement. (**B**) Heart failure.

**Figure 4 ijms-22-10795-f004:**
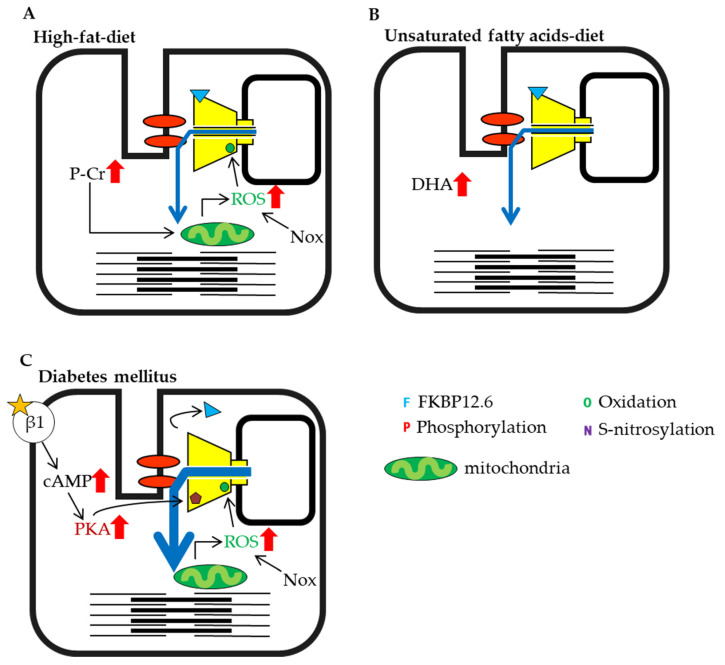
Schematic representation of cardiac ryanodine receptor (RyR2) modulation by intracellular factors under different conditions (part 2). (**A**) High-fat diet. (**B**) Unsaturated fatty acid diet. (**C**) Diabetes mellitus.

**Table 1 ijms-22-10795-t001:** The effects of changes in intracellular factors on the activity of the ryanodine receptor in muscle.

Factor	Normal Level	Stress Level	Effect	Reference
ADP (mmol/L)	0.008	0.15 (high load)	suppression	Lanza, I.R. et al., *J. Physiol.* **2006**, *577*, 353–367
AMP (μmol/L)	0.007	2.7 (high load)	suppression	Lanza, I.R. et al., *J. Physiol.* **2006**, *577*, 353–367
Mg^2+^ (mmol/L)	0.8	1.55 (high load)	suppression	Westerblad, H. et al., *J. Physiol.* **1992**, *453*, 413–434
SR luminal Ca^2+^ (mmol/L)	0.4	0.08 (high load)	suppression	Ziman, A.P. et al., *Biophys. J.* **2010**, *99*, 2705–2714; Rudolf, R. et al., *J. Cell Biol.* **2006**, *173*, 187–193
NO (nmol/L)	50–200	450 (ischemia)	activation	Huk, I. et al., *Br. J. Surg.* **1998**, *85*, 1080–1085
H_2_O_2_ (pmol/min/mg)	100–400	>2 times (contusion)	activation	Bombicino, S.S. et al., *Free Radic. Biol. Med.* **2017**, *112*, 267–276; Goncalves, R.L.S. et al., *J. Biol. Chem.* **2015**, *290*, 209–227; Munro, D. et al., *Free Radic. Biol. Med.* **2016**, *96*, 334–346; Miller, V.J. et al., *Am. J. Physiol.-Endocrinol. Metab.* **2020**, *319*, E995–E1007; Hartmann, D.D. et al., *Free Radic. Res.* **2020**, *54*, 137–149
NADH/NAD ratio	0.04–0.05	>20 times (ischemia)	activation	Kobayashi, K. et al., *J. Mol. Cell. Cardiol.* **1983**, *15*, 369–382; Park, J.W. et al., *Int. J. Cardiol.* **1998**, *65*, 139–147; Zhou, L.; et al., *Am. J. Physiol.-Hear. Circ. Physiol.* **2006**, *291*, H1036–H1046

ADP, adenosine diphosphate; AMP, adenosine monophosphate; ATP, adenosine triphosphate; NAD, nicotinamide adenine dinucleotide; NO, nitric oxide; SR, sarcoplasmic reticulum.

## Data Availability

Not applicable.

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
