# Peer review of "The Ryanodine Receptor as a Sensor for Intracellular Environments in Muscles"

_ijms, 2021, doi:10.3390/ijms221910795_

Round 1

Reviewer 1 Report

The authors proposed a revised manuscript as new submission. After a careful reading of the article, I think it has been substantially improved. Text modifications, additional figures and revised discussion render it suitable for publication.

I have to suggest only few minor corrections, which however do not play my positive evaluation down. If they will be fixed in a reasonable timescale, I support the publication of the Kobayashi et al. work without further delay.

Minor points:

Line 16: “sensor of changes…”

Line 29: “immediately respond to them.”

Lines 282-283: I do not find the experiment where Espinosa et al. showed that ROS scavengers reduced Ca2+ increase after muscle stimulation (Ref. 102). Could the authors carefully retrace and explain their point here?

Lines 301-302: “..suppress channel activity. induced by ATP”,  remove “induced by ATP”

Lines 316-317: “(C) Injury caused by damage heat stress or disuse.” Remove the rest :“Destruction situations caused by injury, heat stress, and dis-use”.

Line 320: “…and are sensitive to external….”

Line 336: “…, to stimulate mitochondrial ROS….”

Line 339: “…mutation in RyR1 display increased…”

Lines 340-341: “…unfortunate enhancement of mitochondrial ROS production, resulting in sudden death”

Line 450: “…similar to the compensatory mechanism occurring in heart failure progress”.

Lines 473-474: “..regulates skeletal and cardiac muscle function.”

Lines 475-476: “ After the occurrence of a critical situation, such as external stress, …”

481-482: “…may be crucial to maintain muscle function…”

Author Response

Reviewer#1

Line 16: “sensor of changes…”
Line 29: “immediately respond to them.”
Lines 301-302: “..suppress channel activity. induced by ATP”,remove “induced by ATP”
Lines 316-317: “(C) Injury caused by damage heat stress or disuse.” Remove the
rest :“Destruction situations caused by injury, heat stress, and dis-use”.
Line 320: “…and are sensitive to external….”
Line 336: “…, to stimulate mitochondrial ROS….”
Line 339: “…mutation in RyR1 display increased…”
Lines 340-341: “…unfortunate enhancement of mitochondrial ROS production,
resulting in sudden death”
Line 450: “…similar to the compensatory mechanism occurring in heart failure
progress”.
Lines 473-474: “..regulates skeletal and cardiac muscle function.”
Lines 475-476: “ After the occurrence of a critical situation, such as external stress, …”
481-482: “…may be crucial to maintain muscle function…”

We corrected indicated sentences according to reviewer’s comments.

Lines 282-283: I do not find the experiment where Espinosa et al. showed that ROS scavengers reduced Ca2+ increase after muscle stimulation (Ref. 102). Could the authors carefully retrace and explain their point here?

We corrected term to “Nox inhibitiors” from “ROS scavengers” based on the experiment of Espinosa et al.. We discussed about the relation between ROS and muscular function based on results of Espinosa et al and Favero et al.

Reviewer 2 Report

The authors have address my concerns adequately. 

The authors may want to consider the following to make the review stronger:

The authors still fail to discuss controversies and unanswered questions concerning the RyR as requested in the previous review.

The review is superficial and would benefit someone very new in the field. This was specified in the first review but not really addressed.  The authors spend just a short paragraph in areas where entire reviews have been written.  If the authors wanted to keep the review short, they could have at least pointed the readers to these reviews.

Minor

Indent line 217

line 304 - "1mmol/L" should be "1 mmol/L"

Author Response

Reviewer#2

Indent line 217
line 304 - "1mmol/L" should be "1 mmol/L"
We corrected indicated sentences according to reviewer’s comments.
We changed the ref. 81 as a more appropriate reference. Therefore, the concentration was changed to 0.1 mmol/L from 1 mmol/L.

This manuscript is a resubmission of an earlier submission. The following is a list of the peer review reports and author responses from that submission.

Round 1

Reviewer 1 Report

The authors extensively revised their original manuscript, answered to all my comments and concerns, and finally applied the indicated modifications in the text and figures. I am completely satisfied of their reviewing work and of their reply letter. However, I found still some errors and few typos that need to be addressed and solved before the final acceptance of the article.

I put the list of my additional comments here below (the line numbering will follow that of the revised version of the manuscript as found in the “ijms-1380183-revisions.pdf” file):

Line 163: “triadin” is the correct noun, not traiadin

Line 181,197,214,227,241,273: adjust and shift by one all the numbers of the chapters since chapter 2.4 was deleted.

Lines 211-212: it will be recommended to list the chapters/sections with the title starting with capital letters and to correct the title of the chapter to the new version: Skeletal muscle injury

Line 212: correct to “…sections of this review”

Lines 238 and 295: correct the title of the “Diet and diabetes” section

Line 289: correct the typo: “acyl-carnitine”

Line 354-357: I suggest to slightly change the title and caption of Fig.2 in order to make it more readable, as follows: “Schematic representation of skeletal muscle ryanodine receptor 1 (RyR1) modulation by intracellular factors under different conditions. (A) Exercise training by eccentric contraction. (B) High-load bearing work and fatigue. (C) Injury condition caused by damage, heat stress or disuse. (D) Aging.”

 Line 364: “…RyR activity does not occur…”

Line 380: “Heat stress and hypoxia…”

Line 382: “…., to increase mitochondrial ROS production (112) and to promote expression…”

Lines 390-394 should be corrected as follows: “Under this condition, NOS is induced by the hypoxia-induced transcription factor HIF-1α  [115] and ROS-production increases with anearobic ATP synthesis, therefore proteins are oxidized [116] and RyR1 binding by FKBP12 is relieved due to the increased ROS and NO ”

Line 397: as I already highlighted in my previous reviewing report, the term “crisis” is not adequate to illustrate the process of muscle decline with age, since aging is not an acute event, but rather a progressive and relative slow process of wasting. I encourage to change the word “crisis” with “critical condition”.

 Lines 435-437: I suggest to slightly change the title of Fig. 3 in order to make it more readable, as follows: “Schematic representation of cardiac ryanodine receptor 2 (RyR2) modulation by intracellular factors under different conditions (part 1). …”

Line 441: “…generation of abnormal Ca2+ waves…”

Lines 510-511: “…The increase of oxidation and the decrease of FKBP12.6 binding to RyR2…

Line 512: correct the typo in “ compensatory”

Lines 522-523: I suggest to slightly change the title of Fig. 4 as follows: “Schematic representation of cardiac ryanodine receptor 2 (RyR2) modulation by intracellular factors under different conditions (part 2). …”

Line 524: “(B) unsaturated fatty acid-diet”

Line 543: “The RyR should not be considered simply as an ion channel, but also as a sensor and receptor that…”

Line 555: correct the typo in “ cardiac”

Reviewer 2 Report

The authors make the claim that the RyR is a sensor of the extracellular environment.  The very superficially discuss many in the intercellular signals that regulate the RyR.  In Table 1, they note how theses intercellular regulators can change, however, these quantities are missing critical references.  For example, the levels they report for SR calcium seem low.  In heart the resting SR calcium concentration is about 1 mM and while it declines during an action potential it is restored.  In fact, in heart during fast beating, it actually increases due to increased time averaged calcium entry through the L-type calcium channel.  Beta adrenergic stimulation increase SR calcium concentration even further.

The author fail to demonstrate how the extracellular environment is “percieved’ by the RyR.  Their examples of changes in aging and disease involve changes in the signaling networks and changes in gene expression which are not direct modulation of the RyR by the extracellular environment.  The claim that the RyR is an extracellular sensor is speculative at best.

Often reviews discuss current controversies and challenges.  However, this review fails to do so.